# Plutonium Signatures in Molten-Salt Reactor Off-Gas Tank and Safeguards Considerations

Nicholas Dunkle [1,*][ID], Alex Wheeler [1], Jarod Richardson [1], Sandra Bogetic [1,*], Ondrej Chvala [1,2][ID] and Steven E. Skutnik [1,3]

1    Department of Nuclear Engineering, University of Tennessee, Knoxville, TN 37996, USA
2    Department of Mechanical Engineering, University of Texas, Austin, TX 78712, USA
3    Oak Ridge National Laboratory, Oak Ridge, TN 37996, USA
*    Correspondence: ndunkle@vols.utk.edu (N.D.); sbogetic@utk.edu (S.B.)

**Abstract:** Fluid-fueled molten-salt reactors (MSRs) are actively being developed by several companies, with plans to deploy them internationally. The current IAEA inspection tools are largely incompatible with the unique design features of liquid fuel MSRs (e.g., the complex fuel chemistry, circulating fuel inventory, bulk accountancy, and high radiation environment). For these reasons, safeguards for MSRs are seen as challenging and require the development of new techniques. This paper proposes one such technique through the observation of the reactor's off-gas. Any reactor design using low-enriched uranium will build up plutonium as the fuel undergoes burnup. Plutonium has different fission product yields than uranium. Therefore, a shift in fission product production is expected with fuel evolution. The passive removal of certain gaseous fission products to the off-gas tank of an MSR provides a valuable opportunity for analysis without significant modifications to the design of the system. Uniquely, due to the gaseous nature of the isotopes, beta particle emissions are available for observation. The ratios of these fission product isotopes can, thus, be traced back to the relative amount and types of fissile isotopes in the core. This proposed technique represents an effective safeguards tool for bulk accountancy which, while avoiding being onerous, could be used in concert with other techniques to meet the IAEA's timeliness goals for the detection of a diversion.

**Keywords:** molten-salt reactors; plutonium signatures; beta particle spectroscopy; safeguards; Off-Gas management

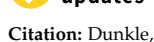



## 1. Introduction

Nuclear reactors produce a tremendous amount of information in the form of radiation. The onus is on researchers to collect, decode, and make use of this information. One of the major goals of this work is utilizing this information to monitor the amount of special nuclear material in the system throughout operation for the purpose of safeguards. Safeguards for conventional nuclear reactors are currently performed with itemized accountancy techniques to maintain the continuity of knowledge on the material at and between designated material balance areas. In these light water reactors, the solid fuel is easily itemized as individual fuel rods with serial numbers. In a molten-salt reactor (MSR), the fuel is in a eutectic mixture with the salt. This fuel salt is fluid and flows throughout the primary loop of the design. While it is possible to use traditional item accountancy techniques for MSR fuel salt before and after its use in the reactor, bulk accountancy techniques must be employed [1]. MSRs challenge the current international safeguards paradigm and encourage the development of new methods. Safeguards for MSRs can potentially utilize non-traditional information sources, such as from radiation within the off-gas tank.

In this paper, we present a conceptual technique to track plutonium buildup in a thermal spectrum molten-salt reactor (MSR) through the analysis of radiation in the off-gas tank. The goal of this novel technique is to observe the isotopes in the off-gas tank and

relate them to the fission reaction in the core via the ratio of isotopes with different fission yields for different fissile isotopes. This paper outlines the methodology and technique with the use of neutronics simulations. Off-gas monitoring has the potential to become a valuable component and defense layer in an overall safeguards plan for an MSR. In an MSR, fission gasses can remain in solution or form bubbles in the liquid fuel salt. Of these fission products, noble gases will reliably form bubbles in the operating fluid. These bubbles can exit the fuel salt via an off-gas tank. The relative production rates of these noble gas isotopes is dependent on the identity of the atoms that underwent fission to produce them. Due to this relationship, the emissions from the off-gas carry information about the amount of different fissile isotopes participating in the reaction in the core. The measurement of gamma rays is possible for both the fuel salt and off-gas. However, the measurement of beta particles is only possible in the off-gas due to its gaseous state.

An analysis on the connected relationship between off-gas and plutonium buildup was performed with simulations in Serpent [2]. The work included in this paper is an expansion of previous work on safeguards considerations for MSRs [1], plutonium signatures in MSRs [3], and also utilizes the NERTHUS neutronics model [4]. These techniques are potentially valuable safeguards and process monitoring tools for MSRs, and thus, deserve further investigation.

MSRs are currently being developed by several companies (e.g., ThorCon Power, Kairos Power, TerraPower, Terrestrial Energy, Copenhagen Atomics, etc.) and, therefore, require novel safeguards techniques. The unique and attractive potential of MSRs is due to their enhanced ability to perform load following [5], their improved passive safety, the fact that they can be used in non-electric energy markets requiring high temperatures, and more. A detailed discussion of the advantages of MSRs is available in [6–8].

## 2. Background

The analysis of the radiation emissions from an object or substance as a means to date it and investigate its composition is a well-understood technique in multiple fields. However, there are unique challenges that make this analysis more difficult in nuclear reactors in general, and molten-salt reactors (MSR) in specific. Firstly, the material structure of the reactor shields any emitted alpha or beta radiation. Secondly, the bulk of gamma and neutron radiation that arrives at a nearby detector would likely overwhelm most detectors. Finally, the operating temperature of a reactor would make cooling any detector difficult or infeasible. These reasons form the basic requirements for any detector to be useful. Despite these challenges, investigations have been conducted on the potential of spectroscopic and chemometric analyses on MSR off-gas [9–11]. The methods performed in that research are focused on the monitoring of iodine in the off-gas via photon interactions. A similar project investigated using gamma spectroscopy on the off-gas [12]. In this paper, we are the first to investigate the potential of using the beta radiation from several krypton and xenon isotopes in the off-gas to conduct process monitoring for the purpose of safeguards. Multi-physics modeling methods have recently been used to simulate MSRs [13]. The MSR model used in this work is the NERTHUS neutronic model made in Serpent [4].

Various noble gas isotopes of xenon and krypton are produced at differing rates from the fission of both uranium and plutonium. This fact has led to the monitoring of noble gas isotopes at nuclear reprocessing facilities to be a tool in the international safeguards toolkit for years [14]. The technique presented in this paper is similar in nature to these methods and represents a novel use specific to MSRs. Unique to MSRs, the special nuclear material and its products are in a liquid state and are circulating throughout the primary loop. As opposed to a reprocessing facility, the special nuclear material in an operating MSR is actively fissioning within the core and its products are decaying all throughout the loop. Due to the active fission reaction, there is a material evolution, which is discussed further in Section 4.2. Through the research performed on the molten-salt reactor experiment (MSRE), much has been known for decades about the behavior and identity of fission products in an MSR [15]. A major characteristic of MSRs is that the noble gases will reliably form

bubbles and exit the salt mixture into an off-gas tank [16]. The foundation of the technique proposed in this paper is that these off-gases can be linked back to the active operation within the reactor and provide information on the production of plutonium.

Plutonium is both produced and consumed within the reactor. Production occurs when $^{238}$U absorbs a neutron, becoming $^{239}$U, and beta decays into $^{239}$Np, and then beta decays again into $^{239}$Pu. Plutonium is also consumed in the reactor by absorbing a thermal neutron and undergoing fission. A further discussion of these phenomena and the overall evolution of the fuel with burnup is included in Section 4.2. The resulting relative participation of $^{239}$Pu and $^{235}$U in the fission reaction through fuel burnup is shown and discussed in Section 5.1.

*Off-Gas Management and Analysis*

The construction and operation of MSRs have multiple waste streams, which take different forms. A discussion of these individual waste streams are available elsewhere [17]. Of particular relevance is the off-gas stream. As an MSR is operated, gases form from within the fuel salt. These gases include reactive gases, residual halides, aerosols, noble gases, and others. Not all gases stay within the fuel salt, and whether they form bubbles or not is dependent on their solubility. Fission product solubility in operating fuel salt is a topic ripe for further research, and analyses have been performed on the topic in the days of the MSRE [15] (p. 54). For the sake of simplicity and applicability to general MSR designs, noble gases were chosen as the focus of this work. In particular, it was shown that krypton and xenon will reliably bubble out of the mixture [18,19]. It may be possible that other gases prove to be useful with this method.

Both the MSRE and the molten-salt breeder reactor (MSBR) [19] (Figure 1) had an off-gas tank in their design to manage gaseous waste streams for disposal. The primary purpose of the off-gas tank was to act as an intermediate storage for the radioactive gases to cool off in before disposal. In this sense, the off-gas system of an MSR is similar to the gas holdup system in conventional light water reactors (LWRs) to deal with their gaseous waste streams. These conventional LWRs, along with the MSRE and the MSBR, also utilize charcoal filtration systems in their designs to remove tritium from the gas [20] (p. 271). In particular, the MSBR planned to hold the off-gas first for two hours before it was vented through particle traps, before being absorbed into charcoal beds for 47 h to remove most of the $^{135}$Xe. In their case, the gas was then to be recycled back into the operating system via the bubble generators [21] (p. 7). MSRs have many of the same benefits as LWRs from the use of an off-gas system, with the significant additional benefit of the removal of gaseous fission products (primarily $^{135}$Xe) during operation. Depending on the location of the MSR plant, it may be required by domestic regulation to contain and/or separate out certain isotopes (e.g., $^{85}$Kr). To comply, an off-gas management system may be required. It is, therefore, reasonable to presume that most MSRs would include an off-gas management system in their design. Research has already been conducted on what form an MSR off-gas management system could be like in a commercial MSR power plant [22] (Figure 4). The example design given includes filtration and separation mechanisms, and the recycling of helium in the system. There has also been additional research on the separation of Xe and Kr produced from a nuclear reactor using metal–organic frameworks [23]. Furthermore, the presence of an off-gas tank receiving gases from an actively fissioning system allows for the ability to analyze the shorter lived (half-life in hours or less) gaseous fission products. This fact places MSRs in a privileged position in comparison to both conventional and advanced reactor designs. Further compounding this benefit is the fact that no intentional processing is needed to begin this analysis because it is done naturally by the MSR. In this sense, an MSR's off-gas system is an ideal place to observe gaseous fission products.

In the MSRE, the off-gas tank was placed downstream from the core in the pump tank, as shown in Figure 1. It is operated using a small pressurizer to maintain the gas pressure in the inlet of the tank and the tank itself. The liquid fuel salt would, thus, continue downstream through the primary loop, while some of the gases would bubble out

into the off-gas tank. The rate of bubbling out is dependent on the specific design of the mechanism and the flow rate of the primary fluid. A design with a more efficient off-gas management system or with one located elsewhere in the primary loop may yield different specific results. However, the relative trends shown in this work can be expected to be representative of most liquid fuel MSR designs. Regardless of the amount of sparging in the system, the viscosity of the salt, or whatever else may affect the overall rate of bubbling out, gases will be produced and exit to the off-gas management system in relative amounts.

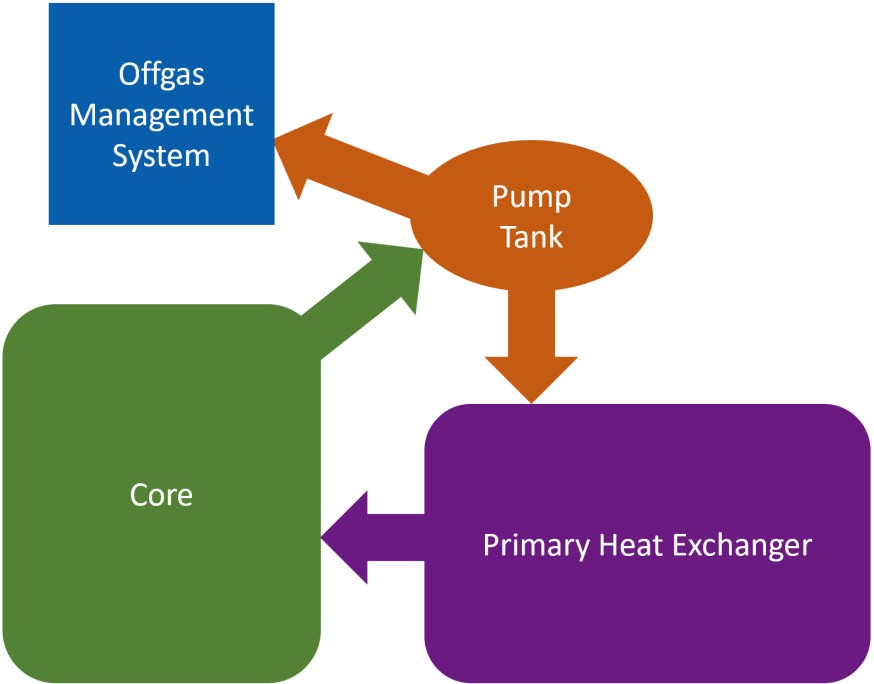

**Figure 1.** Arrangement of the off-gas removal system in relation to the core and primary heat exchanger in the MSRE where arrows indicate flow channels [24].

The major advantage of performing radiation detection at the off-gas tank is that gases have very low self-shielding, which allows for the effective detection of beta particles emitted from them. Normally, with solid fuel or even liquid fuel salt, the fuel is self-shielding and blocks the beta particles from being detected. Therefore, one could place a detector within the off-gas tank itself or within the entryway to the off-gas tank, which is also filled with gas.

## 3. Process Monitoring Using Off-Gas

Process monitoring is a familiar topic to those in nearly any industrial facility, whether nuclear or not. In general, it is the observation of a process for the purpose of maintaining it at a desired state. Such monitoring can take numerous forms in nuclear facilities and for various purposes ranging from control and safety, to plant performance and maintenance. Some examples of process monitoring in nuclear facilities are vibration monitoring, acoustic monitoring, loose parts monitoring, reactor noise analysis, motor electrical signature analysis, and various modeling techniques [25] (p. 2). Process monitoring for the purpose of international safeguards is meant to increase both the timeliness of detection and the amount of information available to identify and verify a possible diversion [26].

Within the topic of process monitoring is on-line monitoring. On-line monitoring (OLM) is process monitoring that is in situ, passive, and in service while the plant is operating. OLM is not necessarily real time, and whether an OLM technique is relatively real time or not is often determined by the data processing time. The off-gas monitoring technique discussed in this paper can be used for the OLM of the reactor and the fissile isotopes within it. While this method focuses on the observation of beta particles, other

methods utilizing infrared spectroscopy have been investigated, and there is potential to incorporate sensor fusion into the design of an off-gas management system [27].

An MSR does not necessarily require an off-gas tank to operate, but neglecting one in the design forfeits some of the advantages an MSR can boast. These advantages include the safeguards and process monitoring uses discussed in this paper, the enhanced load following, and other capabilities [5]. Off-gas tanks are analogous to a holdup tank system used to treat the gaseous waste stream from a conventional light water reactor. Therefore, a system tasked in removing and dealing with the off-gas from an MSR can be expected in most designs. It can, thus, be leveraged as a signal source for monitoring.

The two forms of monitoring proposed in this paper are the tracking of special nuclear material for the purpose of safeguards, and the tracking of isotopes relating to corrosion for the purpose of reactor health monitoring. Enhanced safeguards techniques are a major boon to utilities wishing to construct MSRs internationally where IAEA verification is required. Process monitoring is also a boon to utilities as it allows for data collection on reactor health, which can prevent costly repairs and maintenance downtime [25].

### 3.1. Safeguards Relevance

Safeguards for molten-salt reactors are seen as challenging due to the unique qualities involved in having multiple nuclear systems tightly coupled [28]. The fact that the fissile material is mixed throughout the fuel salt requires the use of bulk accountancy during operation. In addition to this difficulty, the fact that the fuel salt is a homogeneous mixture of actinides, salts, and fission products within a high-temperature and actively fissioning system makes many bulk accountancy techniques, which are used in reprocessing and enrichment nuclear facilities, largely inapplicable.

The use of irradiated fuel measurements has long been a useful non-destructive assay technique for nuclear reactors [29]. However, these measurements have historically been performed on gamma ray emissions from the fission products. The important difference with MSR off-gas monitoring is the utilization of beta particle emissions as a signal source. There are multiple ways in which the information gathered from the off-gas tank can be utilized towards safeguards. The simplest means would be to observe for any disturbances in the trends predicted by the neutronics model of the reactor. A diversion of plutonium from the system would result in a noticeable and timely shift in the fission product ratios shown in Section 5.5. To test this method, diversion simulations can be tested to see how diversions of variable sizes and times would manifest in the fission product ratios. Further development can produce an inference model in which the fission product ratios are compared with the predicted results and are used to determine the material unaccounted for (MUF).

A more advanced technique would be to develop a model that can compare multiple fission product ratios to each other to parse out a fission participation ratio for each of the fissile isotopes. Then, using the known thermal power production of the reactor and the average core flux at that time, the inventory of plutonium, uranium, and the total fissile inventory could be determined. Using this technique would still result in some $\sigma$MUF due to measurement precision. This fact is not unique among bulk facilities, and it can be brought down to a minimum with the proper development of the technique. MSRs would, however, have an opportunity that is unique among bulk accountancy facilities. This opportunity is the use of hybrid accounting techniques within the facility. Specifically, the utilization of item accountancy techniques (serialization, weighing, etc.) before and after use in the reactor. A discussion of this hybrid approach, its specifics, and advantages is found in previous work [1].

### 3.2. System Monitoring

Beyond international safeguards, there are other beneficial reasons for utilities to observe and analyze the fission gases within their off-gas tanks. Such benefits range from monitoring reactor health, with regard to corrosion products detection or the halide

potential of the fuel salt melt ("redox" monitoring), to improvements in the efficiency of the design. These are possible since the rate of gases entering the off-gas is closely connected with certain isotopes with half-lives on the scale of minutes. Those isotopes can, thus, be used as indicators of the reactor condition with only a slight lag.

In particular, $^{139}$Xe has a short half-life of approximately 40 s. Therefore, its equilibrium activity is closely dependent on recent events in the reactor, and it would be a useful isotope for power monitoring. There is also THE observation of $^{137}$Xe, which because of its balanced production between uranium and plutonium, could be used to determine the fission rate in the core.

Measurements related to iodine and xenon are also relevant to potential accident release doses [30]. This is an important research topic since the iodine behavior in molten salts is not well understood. There are indications that iodine release correlates with the fluorine potential of the melt [31]. This would enable both THE monitoring of the salt's fluorine potential and provide confidence for release estimates in MSR safety analyses. An exploration of MSR off-gas monitoring potential for both operational and scientific purposes appears to be an attractive field of future research.

## 4. Methodology

The results completed in this paper were performed by running a depletion simulation of a generic molten-salt reactor in Serpent 2.1.32 [4]. The MSR model, named NERTHUS, is open-source and freely available on GitHub. It was inspired by and shares many similarities with ThorCon's TMSR-500 design [32]. The geometry described in the aforementioned paper is the same used in this project, as shown further in Section 4.1. The only difference as applies to the paper is the fuel cycle. The following are specifically changed in this work: the reprocessing rate, refuel rate, refuel salt, and the initial salt. The initial salt has an enrichment of 2.1%, whereas the refuel salt has an enrichment of 19.75%. Refueling was conducted at a constant rate and with a method developed in previous work [33]. Reprocessing was performed within Serpent using the Bateman equations, characterizing the change in the salt as constants in place of decay constants. The removal of the salt in reprocessing, therefore, acts similarly to decay. The simulation initially started with hot fresh salt at beginning of cycle (BOC) and progressed with small-time steps to better capture the rapid changes. The time steps were set to progressively increase as the system reached a pseudo-steady state, up to a total simulated time of 4 years. In this way, the window of focus was set on the time of most rapid change.

The simulation tracked all of the fission products and their location at each step. In this way, there was a recording in the data file of all of the isotopes within the off-gas tank and their masses, activity, thermal power, and atomic density for each step. The isotope tracking in the off-gas tank was where the data came from for all of the figures used in the results section of this paper. The model considered the fuel salt to be homogeneous. There was no distinction within the model for in-core and out-of-core volume of the fuel salt. Instead, to include the external fuel circulating through the core, the total volume of the fuel salt in the model was larger than that of just the in-core fuel salt. The data library used in the Serpent model were the ENDF/B-VII.0 Evaluated Nuclear Data Library [34]. In this work, the uncertainties of the cross sections and fission yields are neglected. The uncertainty of the fission yields of various isotopes may affect their usefulness as plutonium signatures. Determination of these effects is beyond the scope of this paper, but may be investigated in future work.

### 4.1. NERTHUS Neutronics Model

The model used in this work is the NERTHUS neutronics model. A detailed discussion of the development, methodology and the benchmarks of the model are explained in a previous paper [4]. For reader convenience, the most relevant information about the model is included here. A general overview of the MSR is shown in Table 1. The salt used in this

work was FLiBe salt, which is LiF-BeF$_2$-UF$_4$ (72-16-12 mole%) with the lithium depleted to 99.995% $^7$Li.

**Table 1.** NERTHUS overview.

| | |
|---|---|
| Thermal Power | 557 MW |
| Avg Core Inlet Temp | 617 °C |
| Avg Core Outlet Temp | 682 °C |
| Fuel | Uranium |
| Uranium Enrichment | 2.09% |
| Moderator | Graphite |
| Primary Fluid | LiF-BeF$_2$-ZrF$_4$-UF$_4$ |
| Secondary Fluid | LiF-BeF$_2$ |
| Tertiary Fluid | Hitec Salt [35] |
| Core Residency Time | 15 s |
| PHX Residency Time | 11.6 s |
| Out-of-Core Total Loop Time | 14.6 s |

In the model, the core comprised 84 hexagonal prisms of graphite logs with fuel salt flowing between them. An example of one such prism is shown in Figure 2. Similarly, the full axial cross section of the NERTHUS core is shown in Figure 3 and a longitudinal cross section of the core is shown in Figure 4. There were only four materials used in the NERTHUS neutronics model. These materials were the fuel salt, stainless steel, boron carbide, and graphite. The stainless steel used was SUS-316 with a density of 7.9 $\frac{g}{cm^3}$. The control rods were boron carbide with a density of 2.52 $\frac{g}{cm^3}$ and a boron enrichment of 20% $^{10}$B. The shield surrounding the reflector was made of 90% graphite and 10% boron carbine.

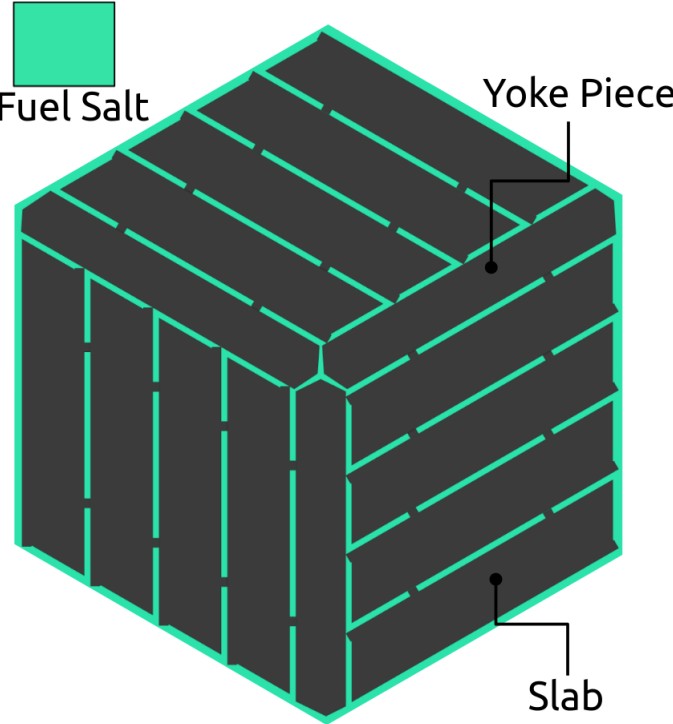

**Figure 2.** An axial cross section of a log used in the NERTHUS reactor.

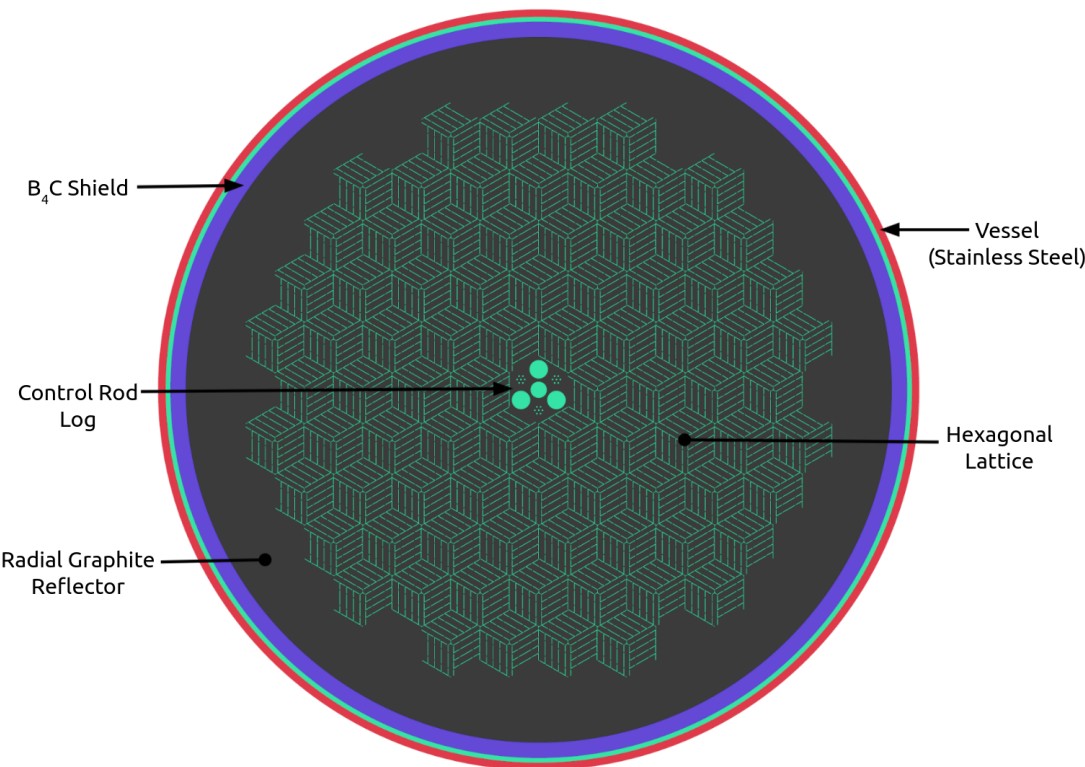

**Figure 3.** An axial cross section of the NERTHUS core.

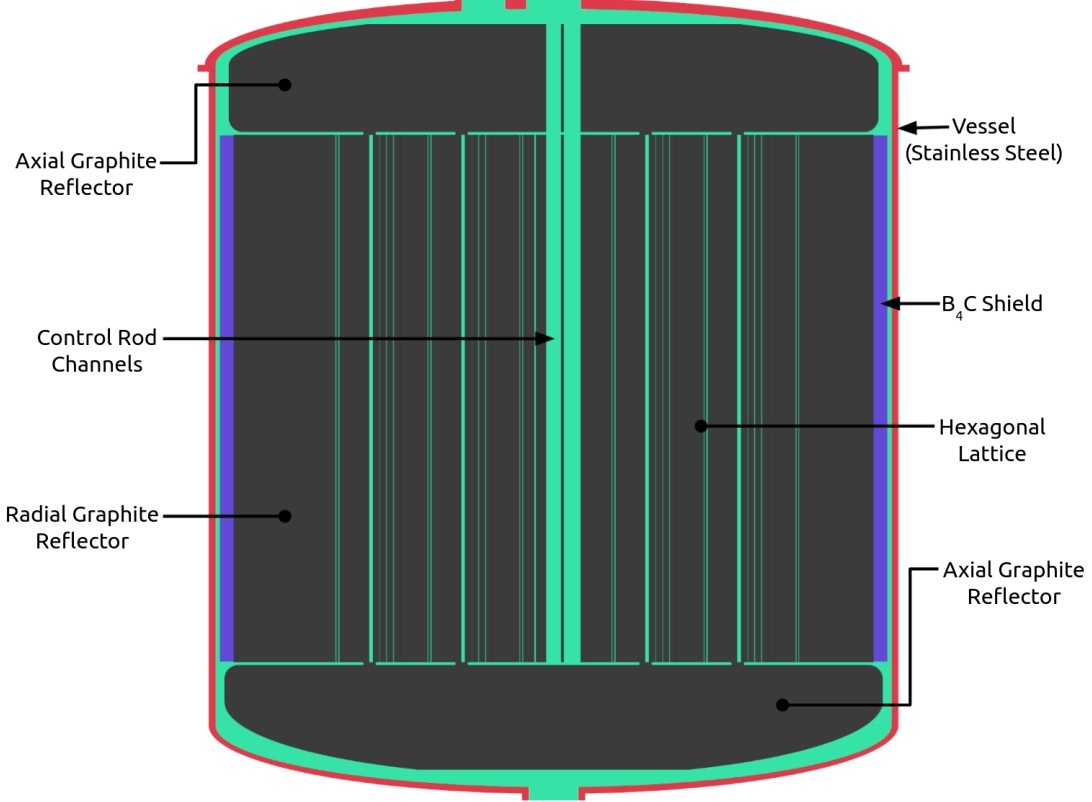

**Figure 4.** A longitudinal cross section of the NERTHUS core with key features labeled.

*4.2. Fuel Evolution*

Operation of a molten-salt reactor (MSR) leads to a complex chemical environment due to the full array of fission products present in "an exotic solvent at a cheerful red heat" [16] (p. 161). Using the MSRE as an example, fresh fuel salt in a generic MSR would be something along the lines of LiF-BeF$_2$-UF$_4$ at a 72-16-12 mole% ratio, with the lithium depleted to 99.995% $^7$Li. Disregarding any slight contaminants from the production process or decays before the start of operation, fresh fuel salt would, thus, be comprised of six isotopes spread across four elements. Specifically, the list of isotopes present would be $^6$Li, $^7$Li, $^9$Be, $^{19}$F, $^{238}$U, and $^{235}$U. At the start of operation, only uranium is fissioning and nearly all of it is $^{235}$U. As a result, a wide array of fission products are immediately introduced to the system. Many of these fission products then decay soon after, creating a cascade of various isotopes into the system. With the sustained neutron flux, many isotopes will absorb a neutron and many of those will have a subsequent decay reaction and change identity once again. Crucially, some of the large quantity of $^{238}$U will undergo this shift: absorbing a neutron, becoming $^{239}$U, beta decaying, becoming $^{239}$Np, beta decaying again, and becoming $^{239}$Pu. The first and most major influential isotope of plutonium to arise is $^{239}$Pu, but as the concentration of it builds up, some $^{239}$Pu absorbs another neutron and becomes $^{240}$Pu. This process repeats up through the more massive plutonium isotopes. Of these, the most influential is $^{241}$Pu, which represents nearly all of the plutonium fission reactions not undergone by $^{239}$Pu. In this way, the variety of elements and isotopes goes from a few at the start of operation to a wide array of isotopes cascading down the periodic table. In general, there are three ways in which fission products can behave. They can either remain soluble in the salt, form insoluble metal particles, which plate out, or form insoluble bubbles, which exit through the off-gas [15,16]. A detailed discussion of the spent fuel salt from an MSR along with further information on how it can be used as sourdough in a new MSR can be found in previous work [33].

The evolution of MSR fuel salt is further complicated through the process of online refueling. Online refueling is an optional design feature that is largely unique among most nuclear reactor designs. One such reactor design that utilized online refueling was the MSBR, which was intended to succeed the MSRE [21]. The process is performed by adding fresh fuel salt into the primary loop to raise the effective enrichment of the bulk fuel salt in the system. To better accomplish this goal, the refueling salt could be at a higher enrichment than the initial fuel salt was. If the enrichment of the initial fuel was 19.75%, then the refueling salt would likely be the same. Conversely, some designs utilize a starting enrichment of approximately 4% for the initial bulk salt, and then refuel with 19.75%. The variance between the enrichment of the initial and refueling salt is, therefore, design-dependent. In tandem, the rate of refueling is also design-dependent. Some designs refuel with small amounts very regularly (or even continuously [36] (p. 20)), while others refuel a large amount infrequently, and again, some designs do not utilize online refueling at all. The act of online refueling has both rapid short-term effects on the fuel salt's isotopic variety and long-term effects on its overall evolution.

Each fissile isotope has a different probabilistic curve of possible fission products. These products each have a yield amount determined by experimental data. The constants used in this work are from ENDF-349 [37]. As discussed in Section 4.2, different fissile isotopes are produced in the fuel over burnup. This phenomenon causes the ratio of these fissile isotopes' participation in fission to evolve over time. We can, therefore, expect the ratio of various fission products actively being produced to change over time as a function of the ratio of the fissile isotopes participating in the chain reactor.

The connection between the fission product production ratio and fission participation ratio is the fission yield of the respective products. A product with a very similar fission yield between the participants would, therefore, be expected to stay mostly constant through burnup once it had reached its saturation point. Conversely, a product with a very different fission yield between the participants would be expected to have its concentration change over burnup to reflect the change in the fissile ratio.

*4.3. Isotopes of Interest*

The variety of fission products is vast. However, there are some considerations that help guide our selection of which isotopes to focus on. Firstly, the isotope must be entering the off-gas tank. To do so, an isotope must reliably be a gas. Which fission products would leave the fuel salt solution to form bubbles in enough quantity is a difficult question. However, we can be reasonably assured that noble gases would exit the solution and persist as a gas [18,19]. Therefore, our selection of isotopes is primarily limited to noble gases.

Next, it is necessary for the isotopes to have a half-life low enough to have a measurable activity but high enough that most of the isotope has a chance to enter the tank. The MSRE had a loop transit time of 25.2 s [20] (p. 102). In the generic MSR model used in this paper, the loop transit time is 30.6 s [4]. In both of these cases, the loop transit time is the time in which a unit of fuel salt takes to complete a journey through the primary loop back to its starting point. In our model, roughly half of that time (15 s) is the core residency time. The off-gas tank is presumed to be attached directly downstream from the core. Therefore, the core-off-gas travel time is a fraction of the total loop transit time (likely only a few seconds), and we can be assured that most of an isotope would have a chance to enter the tank if its half-life is larger than the loop transit time.

Table 2 shows all of the isotopes of interest chosen for this paper. In the table, all presented fission yields are cumulative to better represent the aggregate amount of that isotope ending up in the off-gas tank. However, the use of cumulative fission yields is not without some underlying complexity. Specifically, iodine is a major source of xenon within the system. Therefore, some dragging time for the xenon population should be expected due to the half-lives of its respective parent isotopes being as long as 20 h. This phenomenon is most apparent at the start of the fuel cycle, when the salt is clean and the iodine is not yet in a steady state. Of the chosen isotopes in Table 2, $^{133}$Xe and $^{137}$Xe seem to have potential as a baseline case that has little difference in thermal neutron fission yield between $^{235}$U and $^{239}$Pu. In particular, $^{137}$Xe has such a small difference that it could be used to approximate the number of fissions taking place in the core. Conversely, the krypton isotopes seem to be useful at parsing out the fission participation ratio between $^{235}$U and $^{239}$Pu. In this role, $^{89}$Kr seems to be the best according to an examination due to having the largest fission yield difference in both relative and absolute terms, and a high activity and high energy to release via beta decay. Not included in the table but still important to the discussion is $^{135}$Xe, which is the most significant fission product poison. It can serve as a strong indicator of reactor power. The tracking of $^{135}$Xe and its effect on MSR load following were investigated in previous work [5].

**Table 2.** Half-lives, beta endpoint energies, and fission product yields for $^{235}$U, $^{239}$Pu, and $^{241}$Pu for the isotopes of interest.

| Isotope | Half-Life | $Q_\beta$ (MeV) | $^{235}$U Yield | $^{239}$Pu Yield | Ratio to $^{235}$U | $^{241}$Pu Yield | Ratio to $^{235}$U |
|---|---|---|---|---|---|---|---|
| $^{87}$Kr | 76.30 m | 3.888 | 0.02558 | 0.00989 | 0.3866 | 0.00751 | 0.2936 |
| $^{88}$Kr | 169.5 m | 2.918 | 0.03553 | 0.01272 | 0.3580 | 0.00976 | 0.2747 |
| $^{89}$Kr | 3.150 m | 5.177 | 0.04511 | 0.01453 | 0.3221 | 0.01148 | 0.2545 |
| $^{133}$Xe | 5.280 d | 0.427 | 0.06700 | 0.07016 | 1.0472 | 0.06729 | 1.0043 |
| $^{137}$Xe | 3.818 m | 4.162 | 0.06129 | 0.06010 | 0.980 | 0.06558 | 1.0700 |
| $^{138}$Xe | 14.14 m | 2.915 | 0.06297 | 0.05171 | 0.8212 | 0.06258 | 0.9938 |
| $^{139}$Xe | 39.68 s | 5.057 | 0.05039 | 0.03086 | 0.6124 | 0.04921 | 0.9766 |

## 5. Results

In this section, the results from the simulation are presented. The following are included in the results: the buildup of $^{239}$Pu through burnup, as in Section 5.1; the mass buildup of the isotopes of interest through burnup, as in Section 5.2; the activity present in

the off-gas tank from each of the isotopes through burnup, as in Section 5.3; and the relative activity from each of the isotopes throughout the buildup of $^{239}$Pu, as in Section 5.4. The purpose of presenting these specific results is to demonstrate a clear connection between the presence of plutonium in the core and the amount of radiation of differing energies in the off-gas tank.

### 5.1. Fission Participation Ratio

To determine expectations for the change in fission product production, we must first look at the change in fission participation in the system. Figure 5 shows the relative participation of the primary three fissile isotopes in the fission chain reaction. These isotopes, in order of highest fission rate in the system to lowest, are $^{235}$U, $^{239}$Pu, and $^{241}$Pu. Included in the Pu total curve are all plutonium isotopes undergoing fission in the system. It can be considered that there are three time periods of discussion separated by the two plutonium isotopes reaching a steady state. The first period is from the beginning of cycle to when $^{239}$Pu reaches its maximum fission participation at approximately 400 days. This period starts with only $^{235}$U undergoing fission, but as $^{239}$Pu is transmuted within the system, it begins to increasingly participate in fission. During this time, $^{241}$Pu is also being produced, albeit at a slower rate. It can be seen that $^{239}$Pu reaches saturation nearly 1000 days sooner than $^{241}$Pu. However, as seen in Table 2, $^{241}$Pu has different fission product yields than $^{239}$Pu, which indicates a change in the production of those isotopes should be expected through burnup.

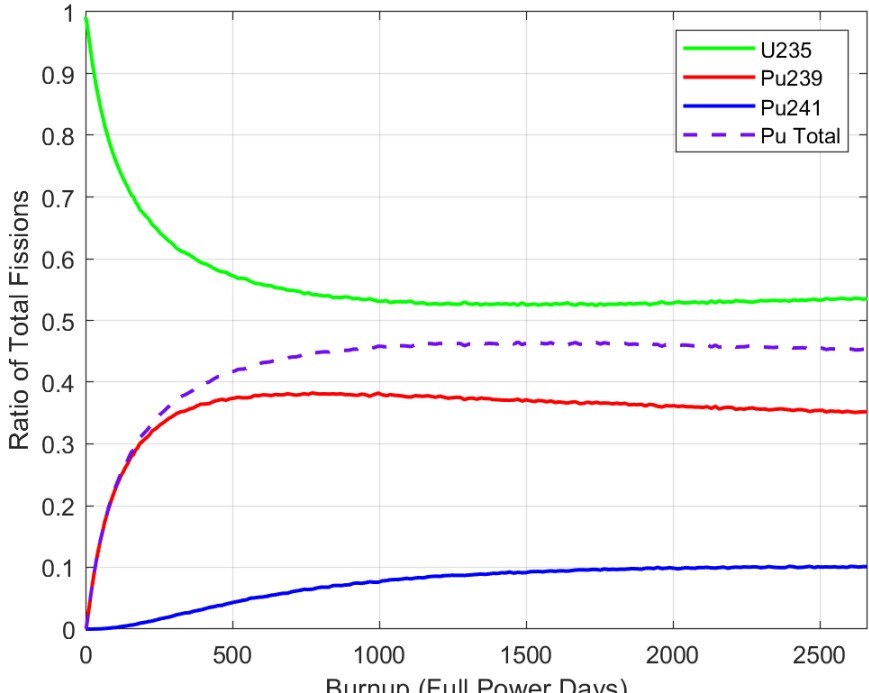

**Figure 5.** Fission participation ratio of major fissile isotopes in the system through burnup when starting with fresh fuel salt. Because we started with fresh fuel, the most rapid change occurs near the beginning of cycle.

The only fissile isotope in the reactor at the beginning of cycle is $^{235}$U. The production of other fissile isotopes is due to transmutation associated with the burnup of the fuel. The relative amount of production is dependent on the design (i.e., breeder vs. burner), but the production method is the same. In any thermal reactor with $^{238}$U, there will be neutron absorptions in $^{238}$U, which turns it into $^{239}$U. From there, $^{239}$U can decay into $^{239}$Np, which can then decay into $^{239}$Pu, a fissile isotope. Therefore, as the reactor is operated, $^{239}$Pu will be produced. We see this in the results as a shift in the Pu/U fission ratio within the first 500

full power days of burnup. The amount of ${}^{241}$Pu is dependent on and will always remain a fraction of ${}^{239}$Pu. This fact is due to ${}^{241}$Pu being produced by the absorption of a neutron in ${}^{239}$Pu turning it into ${}^{240}$Pu and then subsequent neutron absorption. These results are in agreement with those of previous research on the signature of plutonium in an MSR [3].

### 5.2. Mass Buildup

The mass buildup of each of the isotopes in the off-gas tank provides some additional information necessary to make sense of the later results. Figure 6 shows the masses of the krypton isotopes of interest (${}^{87}$Kr, ${}^{88}$Kr, and ${}^{89}$Kr) through burnup, normalized to their maximum value. Similarly, Figure 7 shows the masses of the xenon isotopes of interest (${}^{133}$Xe, ${}^{137}$Xe, ${}^{138}$Xe, and ${}^{139}$Xe). The normalization was performed to more clearly show the relative difference between the isotopes. The maximum mass of the isotopes is also shown. In all cases, excluding ${}^{133}$Xe, the isotope reaches its maximum mass quickly in a matter of hours within the first day of operation. Conversely, ${}^{133}$Xe does not reach its maximum mass until approximately 700 days. The reason for this delay is that the half-life of ${}^{133}$Xe is on the scale of days, whereas the others are on the scale of minutes or seconds, as shown in Table 2. As expected, it can be seen that the inverse is also true: the isotopes with the shortest half-lives reach their maximum values the quickest. With all the isotopes, after the maximum mass has been reached, the mass of the isotope within the tank decreases in connection with the changing fission participation.

In both figures, the insets have sharp changes due to the size of the time steps used in the simulation. Similarly, there is a non-physical disturbance in the data at around 200 full power days also due to the size of the time step changing there. In both cases, this disturbance is an artifact of the simulation process and not a physical phenomenon. The data could be improved with increased computational time involved with smaller time steps nearer to the beginning of the cycle. However, for our discussion here, the important aspect is that the isotopes are shown to quickly reach their maximum values (within 24 h) after the beginning of cycle with fresh fuel salt.

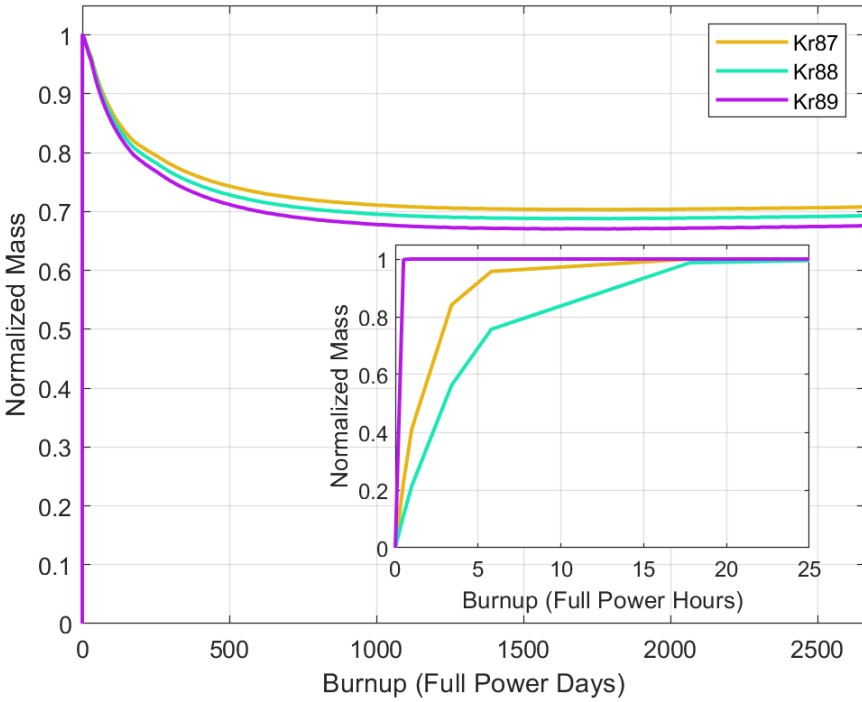

**Figure 6.** The masses of the three krypton isotopes of interest through burnup. They are displayed relative to their maximum mass for better comparison. All three reach their maximum mass within 24 full power hours.

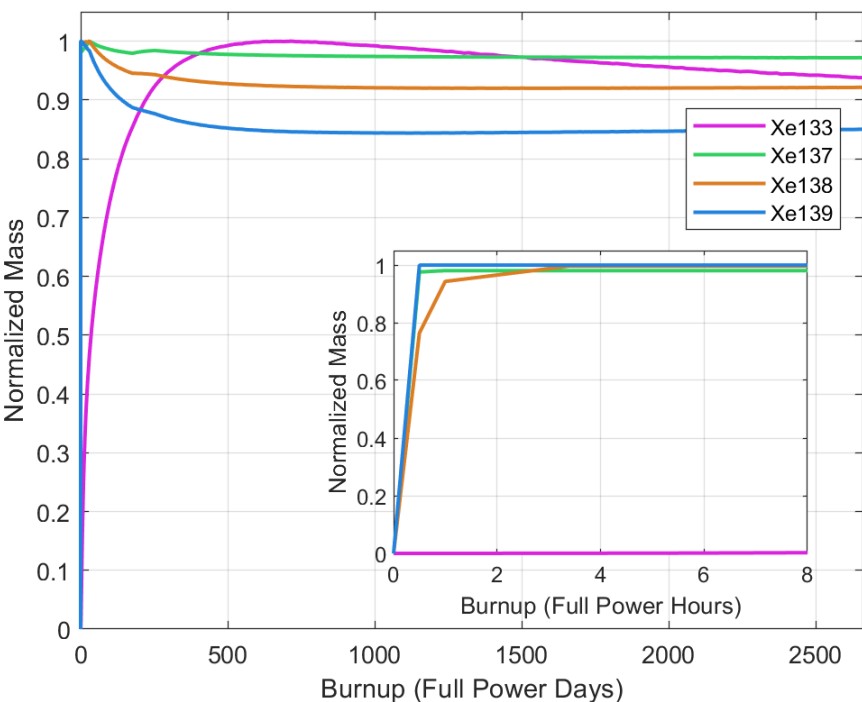

**Figure 7.** The masses of the four xenon isotopes of interest through burnup. They are displayed relative to their maximum mass for better comparison. With exception to $^{133}$Xe, all reach their maximum mass within 24 full power hours.

The maximum mass of each of the seven isotopes of interest is shown in Table 3. An astute reader will quickly see why it was necessary to show the mass buildup of the isotopes relative to their maximum values. To be clear, these masses are the amount within the off-gas tank. The rate of production is comparatively closer between the isotopes than their half-lives, leading to stark differences in the maximum mass. Therefore, as one would expect, the isotope with the largest half-life would have the largest mass, and vice versa. It is for this reason that the maximum mass of $^{139}$Xe is so small in comparison to the others.

An important consideration is that the mass of the isotopes in the off-gas tank is not a one-to-one representation of the activity of those isotopes. As we will see with the activity results, most of the activity from these isotopes is not from the mass, which has already been accrued into the tank, but from the mass that is actively entering the tank. Therefore, tracking the mass of the isotopes of interest in the off-gas tank is indicative of its history, whereas tracking the activity is indicative of the present state of the fission reaction in the core. In addition, this phenomenon suggests that the off-gas tank might not need to be particularly large.

**Table 3.** The maximum mass of each isotope of interest within the off-gas management system through burnup.

| Isotope | Maximum Mass (g) |
|---|---|
| $^{87}$Kr | 0.3864 |
| $^{88}$Kr | 1.2833 |
| $^{89}$Kr | 0.0290 |
| $^{133}$Xe | 13.8307 |
| $^{137}$Xe | 0.0752 |
| $^{138}$Xe | 0.2977 |
| $^{139}$Xe | 0.0085 |

*5.3. Activity*

The activity of each of the isotopes of interest is where the results become particularly relevant. This importance is due to activity being the source of the signal, which would be seen by the detectors. As shown previously, the mass of the isotopes in the off-gas tank is not large (<16 g) due to their short half-lives. However, as shown in Figure 8, the maximum activity of the isotopes is comparatively large. For reference, $10^{17}$ Bq is equal to roughly 2.7 million Curie. The off-gas tank is, therefore, quite hot, radioactively speaking, and there is sufficient signal to measure. If anything, the intensity of the radiation would require special considerations for the detector. However, the specific design of the detector is beyond the scope of this paper and would likely require a further research project on its own. The isotopes of interest chosen for this work have particular usefulness for beta spectroscopy due to the array of different energies they have, as shown in Table 2. There are two pairs of isotopes that may lead to overlapping signals: $^{88}$Kr and $^{138}$Xe, which both have a decay energy of approximately 2.92 MeV; and $^{89}$Kr and $^{139}$Xe, which have a decay energy of 5.177 MeV and 5.057 MeV, respectively. For that reason, isotope pairs must be chosen that have both a difference in fission yield and decay energy, such as the ones shown in Section 5.5.

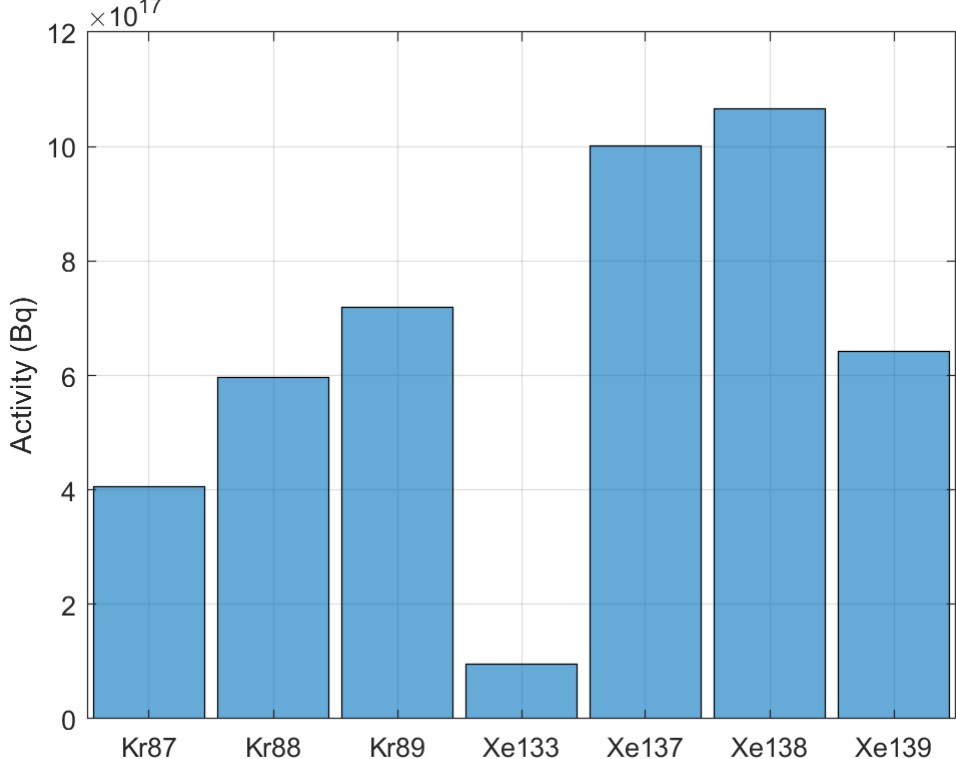

**Figure 8.** The maximum activity of each of the isotopes of interest within the off-gas management system through burnup. Despite reaching the largest maximum mass, $^{133}$Xe has the smallest maximum activity.

The activity in the off-gas tank from each of the krypton and xenon isotopes through the cycle is shown in Figures 9 and 10, respectively. As discussed, all of the isotopes, excluding $^{133}$Xe, reach their respective maximums relatively quickly. However, despite $^{133}$Xe requiring nearly 700 full power days to reach its maximum mass, it only took approximately 200 full power days to reach its activity saturation. This phenomenon is further indicative of the fact that most of the activity observed is from atoms that were recently produced from fission reactions.

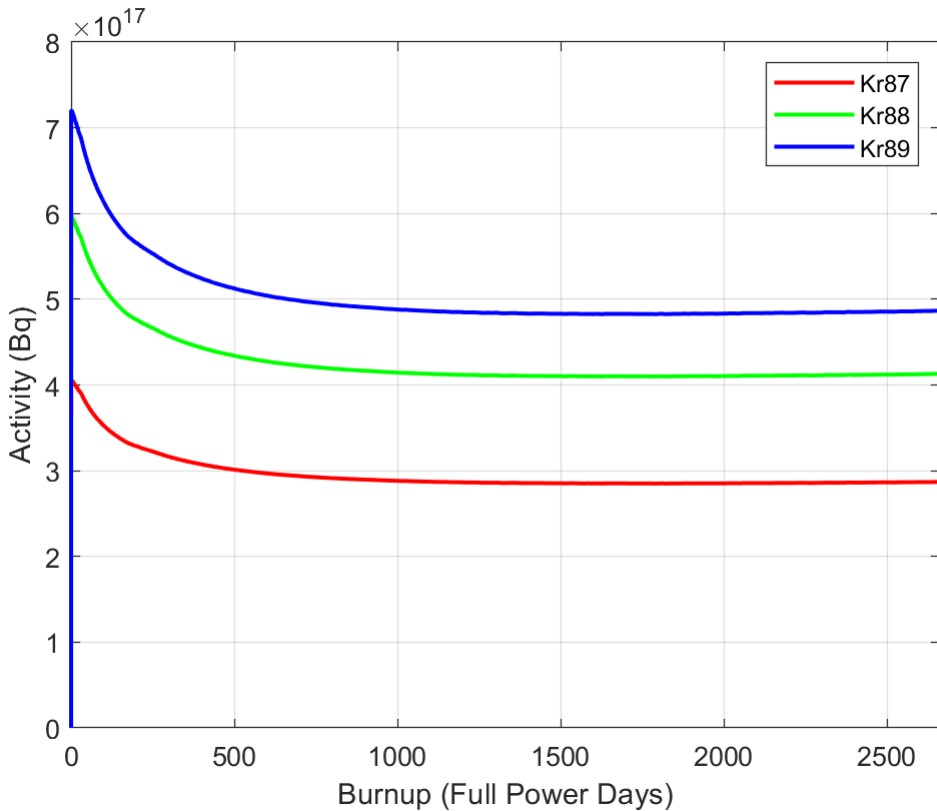

**Figure 9.** The activity of the three krypton isotopes of interest within the off-gas management system through burnup.

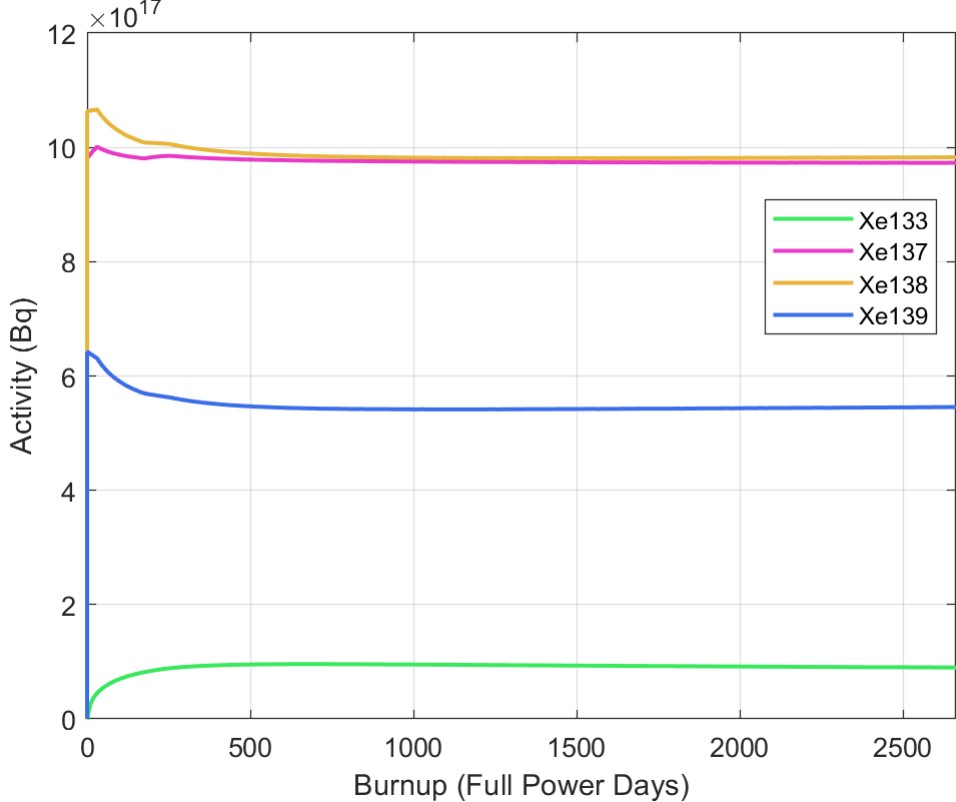

**Figure 10.** The activity of the four xenon isotopes of interest within the off-gas management system through burnup.

### 5.4. Activity Relationship to Fission Participation

Each of the isotopes of interest have a relative shift in steady state activity through burnup due to the evolution of fission participation. This shift in relative activity as a function of fission participation ratio is shown in Figure 11. In the figure, the x-axis is the ratio of all plutonium fissions to all uranium fissions, as discussed in Section 5.1. The isotope activities are normalized to their respective maximums and are the same results discussed in Section 5.3. In this way, by comparing the normalized activity over time to the fission participation ratio over time, the time variable is canceled out and the resulting relationship of isotope activity to fission participation is made clear.

At the highest Pu/U ratios in Figure 11, the activity curves of all of the isotopes loops around and forms a downward trend, with the exclusion of $^{139}$Xe, which has an upward trend. The trends are due to the amount of plutonium reacting in the system decreasing slightly. This decrease makes the same Pu/U fission participation ratio correspond to two different burnup values.

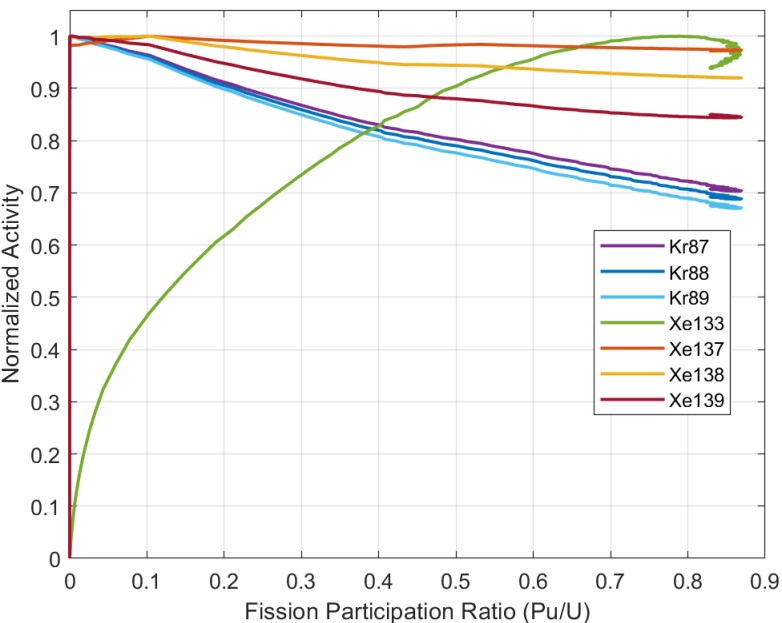

**Figure 11.** The activity of each isotope of interest, relative to the maximum activity that the isotope reached, and compared to the fission participation ratio of plutonium to uranium from Figure 5.

Importantly, the activity of all of the isotopes trends downward with the increased participation of plutonium, with the exception of $^{133}$Xe, which increases. This relationship is to be expected, as seen in the relative fission yields discussed in Table 2. Furthermore, the slope for each of the trend-lines in Figure 11 indicates the degree of usefulness of its respective isotope. For example, $^{137}$Xe is shown to have the least change overall and can, thus, be used as a good partner in ratio with an isotope that changes significantly. As seen in the figure, two such isotopes are $^{89}$Kr, which decreases the most, and $^{133}$Xe, which increases drastically. The relevance of Figure 11 is that it helps indicate which isotopes may be paired well together to connect their ratio to the burnup of the fuel salt and the amount of plutonium in it. These ratios would be used in a method similar to chronometric pairs, but the difference therein is that its connection is to burnup, not directly to time. Some example isotope ratios are provided and discussed further in the next section.

### 5.5. Isotope Pairs

Three isotope pairs were chosen for the demonstration, though others could be used. These pairs are $^{137}$Xe to $^{89}$Kr; $^{133}$Xe to $^{89}$Kr; and $^{133}$Xe to $^{139}$Xe. The results of each are shown in Figures 12–14, respectively. In each, the largest change in ratio occurred within

the first 500 days. This effect was to be expected due to the increase in $^{239}$Pu fissions during that time period, as shown in Figure 5. In this way, the isotope ratios demonstrate a physical connection to the fission participation ratio.

The isotope pair ratios in this section are of the utmost relevance to the off-gas monitoring method proposed in this work. These curves are important because they are what could be expected from an MSR plant that starts with fresh fuel salt and operates through its planned cycle without any outages. The monitoring instrumentation would, thus, observe the rate of beta particle emissions, associate them to each isotope via their beta endpoint energies, as shown in Table 2, calculate the ratios using the chosen pairs, and use the multiple ratios to determine both the burnup and the fraction of fissions belonging to plutonium. From the fission rate known by the reactor's power generation, the amount of plutonium in the system can be assayed. The use of different isotope pairs can decrease uncertainty in the assessment and also allow for the determination of how much of each plutonium isotope there is in the system.

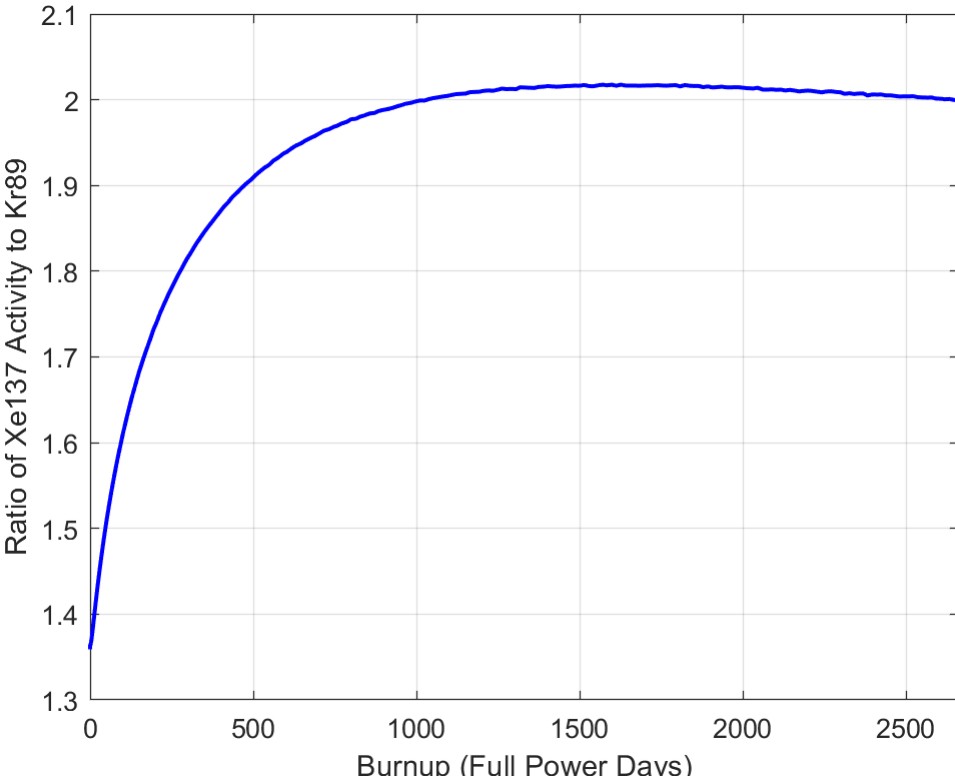

**Figure 12.** The ratio of the absolute activity of $^{137}$Xe to $^{89}$Kr in the off-gas management system through burnup.

Figure 12 shows the activity ratio of $^{137}$Xe to $^{89}$Kr through burnup. These two isotopes were chosen as a pair due to $^{137}$Xe having the least change in relative activity through burnup, and $^{89}$Kr having the most change through burnup, with the exception of $^{133}$Xe. As seen in Figure 11, $^{133}$Xe is a unique case among its peers. The yield of $^{137}$Xe from the fission of $^{239}$Pu is 98% that of $^{235}$U. This rate is the closest of the isotopes of interest to 100% parity. Therefore, $^{137}$Xe can serve as a useful measuring tool for the overall fission rate. In addition, the yield of $^{137}$Xe from the fission of $^{241}$Pu is 107% that of $^{235}$U. This difference allows for $^{137}$Xe to be used to determine the amount of $^{241}$Pu in the system. Similarly, the yield of $^{89}$Kr from the fission of $^{239}$Pu is 32.2% that of $^{235}$U, and from the fission of $^{241}$Pu, it is 25.5% that of $^{235}$U. For these reasons, the $^{137}$Xe to $^{89}$Kr pair possesses the greatest change in its activity ratio across the cycle, demonstrated by the y-axis steps of 0.1.

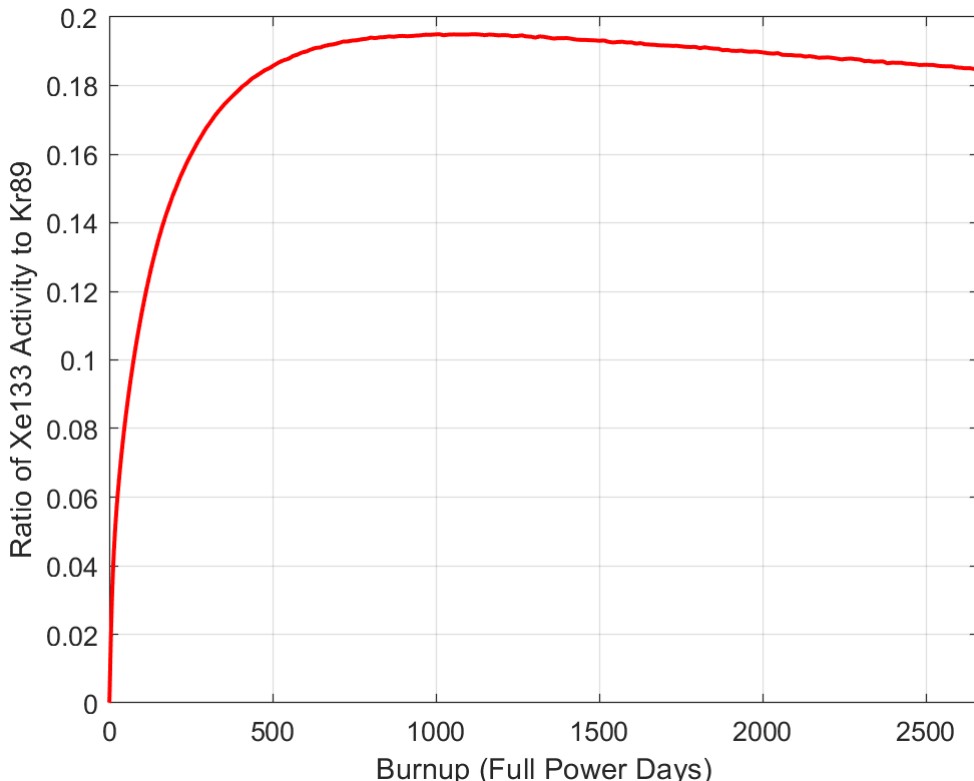

**Figure 13.** The ratio of the absolute activity of $^{133}$Xe to $^{89}$Kr in the off-gas management system through burnup.

Figure 13 shows the activity ratio of $^{133}$Xe to $^{89}$Kr through burnup. As previously mentioned, $^{133}$Xe is a unique case among its peers, and it possesses the greatest change in relative activity across the Pu/U fission participation ratio. For this reason, it was decided to pair it with $^{89}$Kr, which had the second greatest change in relative activity across the Pu/U fission participation ratio. The yield of $^{133}$Xe from the fission of $^{239}$Pu is 104.7% that of $^{235}$U, and from the fission of $^{241}$Pu, it is 100.4% that of $^{235}$U. These values indicate that, like $^{137}$Xe, the activity of $^{133}$Xe is closely tied to both uranium and plutonium. However, due to the longer half-life of $^{133}$Xe and the amount of time it takes to reach its maximum, the signal from it is more indicative of the time from the beginning of the operation rather than the fission rate. The $^{137}$Xe to $^{89}$Kr pair possesses the second greatest change in its activity ratio across the cycle of the three pairs, demonstrated by the y-axis steps of 0.02.

In the previous two figures, $^{133}$Xe and $^{137}$Xe were used to help identify $^{239}$Pu and $^{241}$Pu, respectively. They were each useful in that regard because they have a small difference in fission yield between $^{235}$U and one of the two plutonium isotopes while having a larger difference in fission yield between $^{235}$U and the other plutonium isotope. Specifically, $^{133}$Xe has a 104.7% yield ratio of $^{239}$Pu to $^{235}$U, compared to a 100.4% yield ratio of $^{241}$Pu to $^{235}$U. Conversely, $^{137}$Xe has a 98.0% yield ratio of $^{239}$Pu to $^{235}$U, compared to a 107.0% yield ratio of $^{241}$Pu to $^{235}$U. As shown in Figure 14, the results of this pairing have the smallest ratio of the three isotope pairs.

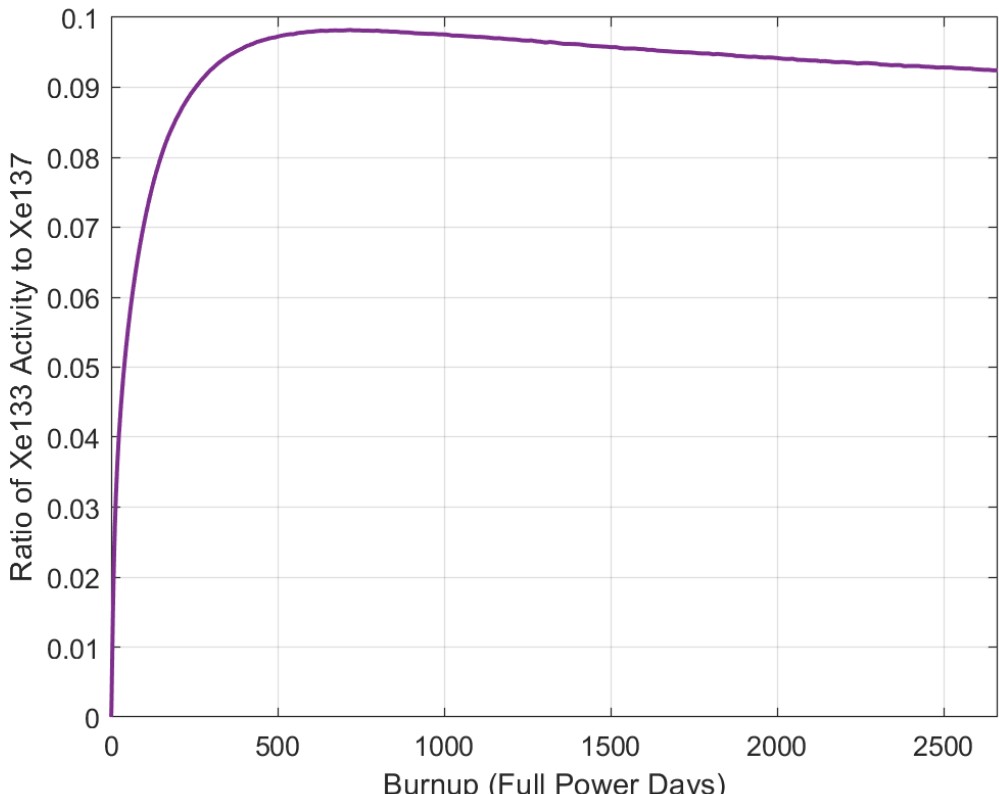

**Figure 14.** The ratio of the absolute activity of $^{133}$Xe to $^{137}$Xe in the off-gas management system through burnup.

If large enough, a diversion of plutonium could be seen in the isotope pairs shown in this section. In future work, the diversion simulations will demonstrate the effect that the removal of plutonium would have on the isotope pairs. As part of that investigation, variable amounts of plutonium could be removed at variable times in the burnup cycle and at variable rates. This process would then be able to demonstrate the overall sensitivity and timeliness of this method to determine whether or not a diversion has taken place. As for how a diversion of plutonium would affect reactor performance, more experimentation and discussion is included in previous work [3].

## 6. Conclusions

Safeguards for MSRs continue to be regarded as difficult due to the inapplicability of conventional techniques used on LWRs. While LWRs can utilize item accountancy solely, the homogeneous nature of the fuel salt in an MSR requires the use of bulk accountancy during operation. This necessity challenges the modern safeguards paradigm to adapt conventional techniques and develop new assay methods. The work presented in this paper is one such novel method, which joins the ranks of various similar studies on the topics of MSR off-gas monitoring, noble gas atmospheric monitoring, and MSR modeling. The intention of this work was to aid in the progression towards the goal of a comprehensive strategy for international safeguards.

The off-gas monitoring technique demonstrated in this paper leverages the unique opportunity MSRs have to measure the beta particle emissions from recently produced fission products in an operating reactor. Similar to other non-destructive assay techniques on irradiated fuel, the underlying principle of the method is that the activity of the fission products can be linked to the identity and amount of fissile isotopes in the system. To demonstrate this connection, the NERTHUS neutronic model [4] was used to simulate a full burnup cycle of a generic MSR and track the isotopes entering the off-gas tank. Seven

were chosen as isotopes of interest due to their 'Goldilocks' half-lives, distinct beta decays, reliable bubble forming, and distribution of fission yields between the three fissile isotopes considered. Of the seven isotopes of interest, three pairs were chosen to demonstrate their connection to the fissile inventory in the system.

**Author Contributions:** Conceptualization, N.D., A.W., J.R., O.C. and S.E.S.; methodology, N.D., A.W., J.R., O.C. and S.E.S.; software, A.W., J.R. and O.C.; validation, N.D., A.W., O.C. and S.E.S.; formal analysis, N.D., O.C. and S.E.S.; investigation, N.D. and O.C.; resources, A.W., O.C. and S.E.S.; data curation, N.D.; writing—original draft preparation, N.D.; writing—review and editing, N.D., A.W., O.C., and S.B.; visualization, N.D. and J.R.; supervision, O.C., S.E.S. and S.B.; project administration, O.C. and S.E.S.; funding acquisition, O.C. and S.E.S. All authors have read and agreed to the published version of the manuscript.

**Funding:** This research and development at the University of Tennessee is being funded by a grant from Oak Ridge National Laboratory (subcontract 4000159472) and from the United States Department of Energy award DE-NE0008793. The authors are grateful for this generous support.

**Conflicts of Interest:** The authors declare no conflict of interest.

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
