# Peer review of "Plutonium Signatures in Molten-Salt Reactor Off-Gas Tank and Safeguards Considerations"

_jne, doi:10.3390/jne4020028_

Round 1

Reviewer 1 Report

The focus of this paper is on MSR safeguards, and the idea of tracing noble gas isotopic ratios (leveraging gaseous nature and beta particle emissions) back to the relative amount and types of fissile isotopes in the core. This is of high interest to the safeguarding community, and important to be considered and utilized as MSRs are developed. However, it’s unclear if there has been any other work on this specific idea/feasibility previously published—this should be addressed; if there has, reference it, and if not, say this is the first such work.

Overall, the communication of this idea and results needs a fair amount of improvement. The list below includes: fixing minor typos, rephrasing some moderate grammatical errors throughout, clarifying the presentation of plots/data, and a number of requests for more scientific content, comment/analysis, and references. These suggestions are made in order to elevate the quality of the paper and if made, will bring the paper into an acceptable form for publication.

·       Most of Section 2.1 is not needed at all and should be struck from the paper. The general and not very thorough description of an MSR (only one design of MSR, by the way) is not adding anything to the actual work of the paper or the reader’s understanding of the data and analysis of results, and is done better elsewhere—which is mentioned as “additional reading available” on page 4 and with (only one!) reference, #13 (why not more references on the first sentence of this paragraph? Especially from authors from ORNL?). A better, more complete definition of the “generic MSR” used for the modeling really must be done, instead, in the Methodology section.

·       To the above point: What *exactly* is the “generic MSR” used in this work? There is some type of hint in the final sentence of section 2.1 that it is MSRE’s FLiBe salt/fuel composition and design type: “…the MSRE’s topology and design is seen as the generic case for an MSR…” but this is not at all clear if this is meant as a general comment for the field (which, if this is what was meant, is not true) or if this is meaning *for this paper*.

·       In the first paragraph of section 2.2, there is mention of solubility, and how not much is known. Then it’s stated that “for simplicity”, noble gases are chosen for this work. But, is there anything known/published about the behavior of noble gases in salt (one would assume so, from the MSRE?), and if so why is nothing referenced here to support this choice?

·       Page 5. First sentence of the second paragraph of section 2.2 contains the acronym MSBR without definition. This is the first mention of MSBR in the paper. Later, on page 7, it is defined. Move definition to the first mention on page 5 and delete definition on page 7.

·       Page 6, right above Figure 1: “Therefore” is capitalized erroneously.

·       Figure 1 on page 6: need an improved quality image overall. This is not high enough resolution (it’s very blurry), and the label “OFFGAS TANK” is unprofessional looking in choice of font; also, why not label other parts of the image?

·       The paragraph above Figure 1 on page 6 includes: “rate of bubbling out is dependent on the specific design of the mechanism and the flow rate of the primary fluid”. This is a crucial part of the feasibility of this proposed measurement. There are gaps in the knowledge/accurate data about the fundamental properties of these systems (viscosity, etc.). This point isn’t discussed or taken into account. Needs to be addressed, mentioned as potentially affecting the results (or if not, explain why this is not an expectation).

·       Page 6. The sentence starting with the word “Specifically” in the first paragraph of Section 3 is not a complete sentence.

·       Page 6. The Methodology section is lacking a basic explanation of the system being modeled. It’s mentioned that it “shares similarities” with ThorCon design, but why isn’t there just a straightforward description of the fuel salt? That is, solvent salt, the fuel / fissile material identity/components, the concentrations, etc.?

·       Page 7. To the above point, in the section 3.1, there is a description of “a generic MSR” fuel salt, but even that is unclear in presentation – it states “…would be something along the lines of…” and describes a FLiBe system with uranium—is this the system being studied herein? Unclear.

·       Page 7. The sentence starting with “Many of which” in the middle of the first paragraph of section 3.1 is not a complete sentence.

·       Page 7. No hyphen needed in “plate-out”.

·       Please rephrase the many uses of starting a sentence with phrases like “Of which”—one example is near the top of Page 8—this is incorrect grammar. Similar instances found all throughout paper, Page 17 top “In which,” and “Whereas” are just two, another is “Beyond which” on page 21

·       A word seems to be missing from the first sentence of the first complete paragraph on page 8: “…a very important _____ innate to the evolution of…”

·       To the above point though, this paragraph (starting with “For our discussion…” on Page 8) can just be completely struck from the paper, it is redundant and therefore unnecessary.

·       Page 8. Punctuation missing after ENDF-349 (perhaps the comma is meant to be a period?)

·       Page 8, in Isotopes of Interest section, again mention of “generic MSR used in this paper” – but what is this generic MSR, and how (if at all) is it different than the MSRE?

·       First paragraph of section 3.2: again mention is made re: the choice of noble gases, their solubililty/”reliably bubbling from salt” but there are zero references—please add references here, or make mention to why/reasoning for making the statement including the word “reliably”

·       Table 1 caption: make mention in the caption that these ARE the isotopes of interest chosen for this work

·       Page 9. First sentence: should be “are the results” not “is the results”. Also, better to rephrase both the first sentences; the passive voice is awkward.

·       The sentences starting “There are three isotopes…” and “The specific reasoning…” are unnecessary and should be struck.

·       Page 10+. Please include a discussion as to what exists in the literature to date with regards to fission participation/any burnup data as is presented in this work, for molten salts, for noble gases. If this is the first ever of this type of study and there are no references to point to, please make mention of this.

·       Page 11. The size of the time steps/sharp changes in the data are noticeable, and the mention/discussion is appreciated. Is this something that is a limitation of the modeling? Is it a result of limited computational time/availability? Is it possible to improve this data with more computational time and/or an improved model?

·       On all plots: Fix captions: All captions need to be much more descriptive. Just saying things like “Xenon masses” is insufficient. Also: there are titles *and* captions on every plot. This is redundant, distracting – suggestion is to just delete titles and add the titles’ content to the captions (but also improve captions beyond just this)

·       “beyond the scope of this paper” is used six times in this manuscript. At least one of these can be struck—one good candidate for deletion is on page 12.

·       Figure 5 needs to be replotted/presented in an improved way. The Xe133 data swamps out all other data and the plot is not useful. Adjust the y axis or otherwise change the presentation of this data. Kr89 and Xe139 data is not even visible/readable.

·       End of first paragraph of section 4.3: change “it” to “its”.

·       Page 13, strike “for the interest of the reader” – this is unnecessary.

·       Section 4.3, page 13, there’s a mention of the detector itself, but it is quickly stated that detector design is not included in this work. But this paper’s focus is all about detection / a detection method, and a discussion of feasibility is crucial here. Does the technology even exist/ is anything capable now to detect the resulting differences /amounts determined in this paper’s main work? It’s mentioned there would be “plenty of signal” but what about sensitivity, resolution, signal to noise, thresholds to detect diversion or other actual changes to healthy/nominal reactor operation? It’s fine to say the details are elsewhere in the literature, but a comment or discussion or references must be included at minimum since this is so key to the whole point of the paper.

·       Page 20. “Both 133Xe and 137Xe have been shown to be interesting. For this reason…” Please rephrase this point/ sentences and add meaning. What does this mean? Interesting how, why?

·       There is a typo or a missed editing mistake in the sentence beginning “With the highest yield…” top of page 20.

·       The entire section 5 is misplaced within the paper. It belongs in the beginning portion of the paper, not near the end.

·       Page 21, OLM is defined and then the acronym is used in the rest of the paragraph except in one instance mid-paragraph, where the full definition is used again—just delete that and substitute the acronym

·       Page 21, very bottom, rephrase sentence beginning with “There are multiple ways…”, it contains an error.

·       Page 22, top. Why wasn’t a diversion simulation done? Is this very challenging? A theoretical example of how this detection method would actually play out would add great value to the paper; if too challenging/time-intensive to do such a simulation, even a calculation and explanation outlined as to what a specific ratio change would look like and then actually mean in terms of diversion or reactor performance anomaly would be very useful in this paper.

·       No need to define acronyms at the end of a paper—remove from Conclusion.

Reviewer 2 Report

The authors present a very interesting work dedicated to process controlling and safeguarding Small Modular Reactors by means of the offgas signature. This work is of utmost relevance because existing process control and safeguards measures from LWR cannot be transferred to the much SMR system which seems to be much more complex.

While I was reading and enjoying the interesting and exciting manuscript, I identified a very little number of points that I want to address which might lead to a little improvement of the excellent paper from my point of view:

-      E.g, on page 6, in the second paragraph the authors inform that “the rate of bubbling out is dependent on the specific design of the mechanism and the flow rate of the primary fluid.”. I am wondering if these input parameters are usually accessible or will be provided by the operator? I guess that is essentially needed to safeguard the SMRs if I am right.

-      Page 7, second paragraph the authors mention that “the model considers the fuel salt to be homogeneous.” What is definitely known about the homogeneity of the fuel salt

-      My next comment is dedicated to the figures’ captions. I guess it would be very helpful to add more information. In Figure 2, 7, 8 “across burnup” could be added similar to Figure 10.

Other propositions are

Figure 3: Krypton masses in offgas tank across burnup

Figure 4: Xenon masses in offgas tank across burnup

Figure 5: Maximum masses of isotopes of interest

Figure 6: Maximum activity from isotopes of interest

-      Page 5, Chapter 2.2, first line of second paragraph: Please, define MSBR. I guess the abbreviation is the first time mentioned here and needs to be defined.

Finally, two typos I want to mention.

-      Page 6, second paragraph, last sentence: “…and can therefore…”

-      Page 8, third paragraph, third line: A dot is missing: “… ENDF-349. [23] As discussed …”

To conclude: The authors present here an excellent work with substantial results providing important input to the process control and safeguards relevance of offgas monitoring of SMRs. I recommend publishing these results in Journal of Nuclear Engineering after minor revision.
